# Interaction between Sulfate and Selenate in Tetraploid Wheat (*Triticum turgidum* L.) Genotypes

**DOI:** 10.3390/ijms24065443

**Published:** 2023-03-13

**Authors:** Eleonora Coppa, Silvia Celletti, Francesco Sestili, Tanja Mimmo, Maria Dolores Garcia Molina, Stefano Cesco, Stefania Astolfi

**Affiliations:** 1Department of Agricultural and Forest Sciences (DAFNE), University of Tuscia, Via S. C. de Lellis, 01100 Viterbo, Italy; 2Faculty of Science and Technology, Free University of Bozen, 39100 Bolzano, Italy

**Keywords:** biofortification, nutrient interaction, selenium, sulfur, wheat

## Abstract

Selenium (Se) is an essential micronutrient of fundamental importance to human health and the main Se source is from plant-derived foods. Plants mainly take up Se as selenate (SeO_4_^2−^), through the root sulfate transport system, because of their chemical similarity. The aims of this study were (1) to characterize the interaction between Se and S during the root uptake process, by measuring the expression of genes coding for high-affinity sulfate transporters and (2) to explore the possibility of increasing plant capability to take up Se by modulating S availability in the growth medium. We selected different tetraploid wheat genotypes as model plants, including a modern genotype, Svevo (*Triticum turgidum* ssp. *durum*), and three ancient Khorasan wheats, Kamut, Turanicum 21, and Etrusco (*Triticum turgidum* ssp. *turanicum)*. The plants were cultivated hydroponically for 20 days in the presence of two sulfate levels, adequate (S = 1.2 mM) and limiting (L = 0.06 mM), and three selenate levels (0, 10, 50 μM). Our findings clearly showed the differential expression of genes encoding the two high-affinity transporters (*TdSultr1.1* and *TdSultr1.3*), which are involved in the primary uptake of sulfate from the rhizosphere. Interestingly, Se accumulation in shoots was higher when S was limited in the nutrient solution.

## 1. Introduction

Selenium (Se) is an essential nutrient for metabolism in humans with a recommended dose of 55 µg per day [1,2]. It is involved in the protection of the thyroid, plays a key role in the immune system, and is also an antioxidant, thus its deficiency causes diseases related to increased free radicals, such as premature aging, chronic inflammatory, and degenerative diseases [3]. Although Se has been quite neglected in comparison to iron, zinc, and iodine, it is worth highlighting that Se deficiency is not rare with recent estimates provided evidence that it is a nutritional disorder affecting over 1 billion people in the world [4]. Considering that the main Se source for humans is from plant-derived foods, enhancing Se concentration in plants might be a crucial approach to increase daily Se intake. However, Se is not considered an essential element to plants, even if beneficial effects of this trace element have been observed in plant growth and development [5,6], and Se content in plant tissues is closely related to its availability in soil and a plant’s ability to take up it [7].

Selenium exists in four oxidation states (−2 selenide, 0 elemental Se, +4 selenite, and +6 selenate) [8] and commonly its concentration in most soils is very low. Only in some defined areas (seleniferous soils) is its concentration sufficiently high to be toxic [9]. Among the different forms, selenate (SeO_4_^2−^) is the most soluble form in alkaline and well-oxidized soils. Selenate uptake occurs through the root sulfate transport system, because of their chemical similarity [4,10]. From our current understanding, root uptake systems include high-affinity sulfate transporters and the selectivity of plant transport toward selenate and sulfate seems to be strongly associated with either the S availability in the growth medium or the plant S nutritional status. Sulfate availability in the rhizosphere generally inhibits selenate uptake, whereas S deficiency conditions lead to the upregulation of sulfate transporters (SULTRs) commonly associated with a higher uptake of Se [11]. From our current understanding, selenate uptake mainly occurs through SULTR1;2, while SULTR1;1 is involved in selenate uptake under S deficiency [4]. Once inside, selenate is translocated to the shoot by the transporter of group 2 SULTR2;1, and it is postulated that the transport in the chloroplast could be mediated by the SULTR3;1 transporter. Moreover, it should be highlighted that, besides selenate, selenite and other Se compounds may be acquired by plants [12], which may or may not have the ability to accumulate Se. Accordingly, plants can be distinguished into Se non-accumulators and accumulators [13] and, in the latter, this attribute is due to their ability to discriminate between Se and S, preferentially acquiring the first when the second is also present [14].

Concerning the assimilative pathway, it closely resembles that of sulfate with the reduction and organic incorporation steps taking place mainly in leaves [8]. The first enzyme in the sulfate assimilation pathway is adenosine triphosphate sulfurylase (ATPS), which catalyzes the reaction of sulfate activation leading to the production of adenosine 5′-phosphosulfate (APS). Likewise, selenate is activated by the same enzyme with the formation of adenosine 5′-phosphoselenate (APSe), followed by reduction to selenite by APS-reductase, and then the reduction of selenite to selenide using glutathione (GSH) as the reductant. Selenide is finally incorporated into selenocysteine and selenomethionine [4,13]. In this regard, it is important to remember that in non-accumulating plants, the substitution of cysteine and methionine in proteins with selenocysteine and selenomethionine could cause a loss of function in these proteins with negative effects for plants [13].

The evidence so far provided suggests that the relationship between S and Se could be exploited as an important approach to increase Se concentration in the edible part of plants, possibly associated with the selection of crop genotypes possessing a more pronounced ability to accumulate Se [15], with beneficial effects for human health.

Recently, efforts for the biofortification of food crops have attracted much attention. An interesting crop is represented by wheat, the second most cultivated crop worldwide, whose grain is also the most consumed cereal [16,17,18]. Interestingly, in wheat, Se is mostly present as selenomethionine (up to 90%), highly bioavailable from a nutritional point of view, in contrast to the inorganic forms, such as selenate, which are less bioavailable [17,19,20].

The aims of this study were (1) to characterize the interaction between Se and S during the root uptake process, by measuring the expression of genes coding for high-affinity sulfate transporters and (2) to explore the possibility to increase plant capability to take up Se by modulating S availability in the growth medium. Four different tetraploid wheat genotypes were selected as model plants: a modern genotype, Svevo (*Triticum turgidum* ssp. *durum*), and three ancient ones belonging to *Triticum turgidum* ssp. *turanicum*, namely Kamut^®^, Turanicum 21 (PI184543), and Etrusco.

## 2. Results

### 2.1. Biomass Production and Chlorophyll Content

Plants grew healthily under all treatments, without any significant symptoms of damage during the experimental period but showed significant differences in biomass yield due to the different treatments (Figure 1).

Kamut plants showed better growth and biomass production, at both shoot and root levels, under all conditions, as compared to the other genotypes (Figure 1).

Low S supply in the nutrient solution (L condition) caused a significant reduction in shoot biomass only in Svevo and Kamut plants grown in the presence of 50 µM Se (−40 and −20% with respect to their relative S-sufficient control, respectively) (Figure 1A). On the other hand, a stimulatory effect from low S supply on shoot growth was also present in both Svevo and Kamut plants. In particular, shoot growth increased by 90% in Svevo plants grown without Se, and by 30% in Kamut plants grown in the lowest Se dose (10 µM) (Figure 1A).

The effect of Se addition to the nutrient solution was not only dependent on genotype, but also on S supply. In both Tur-21 and Etrusco plants grown in S-sufficient media (S condition), Se addition did not significantly affect shoot development (Figure 1A). Increasing the Se concentration from 0 to 10 µM increased the shoot biomass of Svevo plants (+60% compared with control plants), whereas it was slightly decreased in Kamut (−15% compared with control plants) (Figure 1A). In the latter genotype, a slight shoot growth stimulation (+15% compared with its relative control) was also found by adding 50 µM Se to the nutrient solution (Figure 1A).

On the other hand, Se addition to a low S medium was shown to negatively affect the shoot growth of all tested genotypes. In Svevo plants, shoot growth decreased progressively when increasing Se concentration in the medium (−15 and −65% from 10 to 50 µM Se), whereas only the highest Se dose (50 µM) prevented shoot development in the other genotypes (Figure 1A).

In all genotypes, the largest root fresh biomass was obtained with plants grown at the lowest S supply (L condition) and without Se (L0), but the effect of the treatment was also dependent on the wheat genotype (Figure 1A). In particular, the development of root apparatus was increased to a larger extent in Svevo and Etrusco plants (2.5- and 2-fold higher than control, respectively), and to a lesser extent in Tur-21 and Kamut plants (47 and 23% higher than control, respectively). Surprisingly, the presence of 10 µM Se in the growth medium cancelled out the effect of S limitation (L) on root growth in all tested genotypes, except for Etrusco, in which the stimulation induced by the low S supply was still evident (+30% with respect to its relative control) (Figure 1A). More surprisingly, our results showed that 50 µM Se completely reversed the low S-induced stimulation of root growth. The results showed that the root biomass accumulation of all tested genotypes was prevented by the presence of Se, and the inhibition ranged from 10% in Tur-21 to 50% in Etrusco (Figure 1A).

Leaf greenness (SPAD) changed with the nutritional treatments (Figure 2). Low S availability without Se (L0) slightly but significantly reduced chlorophyll content by 14, 15, 6, and 10% in Svevo, Tur-21, Kamut, and Etrusco, respectively (Figure 2). Leaf greenness progressively decreased in low S (L) Kamut plants with an increasing Se concentration in the growth medium (compared to control, by 9% and by 15% with 10 and 50 µM Se, respectively), whereas in all the other genotypes, SPAD values slightly increased with the addition of 10 µM Se to the low S medium (by 4, 10, and 6% in Svevo, Tur-21, and Etrusco, respectively), to later decrease towards the control (L0) values by adding 50 µM Se (Figure 2). Concerning the sufficient S condition, SPAD values showed a progressive decrease with an increasing Se concentration in the nutrient solution for all the genotypes except Etrusco, in which chlorophyll content was not affected by the Se treatment (Figure 2). The highest inhibitory effects on SPAD values were observed in Svevo and Kamut plants following the addition of 50 µM Se (compared to control, by 13 and 14%, respectively) (Figure 2).

### 2.2. Sulfur and Selenium Concentrations

The total S concentrations, on a dry weight basis, in the shoots and roots of different wheat genotypes are shown in Table 1 and Table 2. The total S concentration was affected by S supply but also by Se addition, although in different ways in the four genotypes studied.

The S concentration in the plants grown under low S conditions (L) was generally lower, or at least not significantly different than that measured in the control ones (S), for most genotypes and Se treatments. However, some exceptions were found in both shoot and root tissues. In particular, the S concentration of L plants was significantly higher than the S controls in both the shoots and roots of Svevo plants grown with 10µM Se (1.5- and 3-fold higher, respectively) (Table 2), in the shoots of Tur-21 plants grown without and with 50µM Se (2- and 1.5-fold higher, respectively) (Table 2), and in the shoots of Kamut plants grown without Se. Surprisingly, in these latter ones, the total S concentration in the L0 condition was more than four-fold higher than those found in the S0 control (Table 1 and Table 2).

By comparing the accumulation pattern of total S as a function of Se supply, we found that it could vary depending on genotype. In the shoots of all genotypes, except Kamut, grown in S condition, S accumulation was increased by the addition of Se (Table 1); roots share the same pattern, including Kamut (Figure 3B). As regards the L condition, the S concentration in shoots was increased by the Se addition, but only in Svevo and Etrusco genotypes, at the lowest (10 µM) and highest (50 µM) dosage, respectively, whereas in Tur-21 and Kamut, it decreased with the addition of Se (Table 2); in roots, S accumulation was only increased by the Se supply in Svevo (only 10 µM) and Tur-21 plants, whereas no significant differences were found in both Kamut and Etrusco (Table 2).

The root sulfate transport system is not only responsible for sulfate uptake but also for selenate uptake, because of their chemical similarity [4,10]. For this reason, the role of S nutrition on Se accumulation in the root and shoot tissues of wheat plants was determined, and the results obtained are reported in Table 1 and Table 2.

With a few exceptions, as expected, when plants were grown with an adequate S supply (control condition, S), there were no significant differences in the Se concentration in the considered genotypes in both below- and above-ground biomass, with or without adding Se (Table 1). The only exception was observed for Svevo and Tur-21 plants, which exhibited an increased accumulation of Se in root tissues when grown with additional Se in high and low concentrations, respectively (Table 1). On the other hand, a low S supply (L) clearly stimulated Se accumulation in wheat genotypes. In particular, in shoot tissues, Se concentration increased with the increasing Se supply in the growth medium in all genotypes (Table 2). A similar pattern was also observed at the root level, except for Tur-21 plants, in which the Se supply did not change Se accumulation in root tissues (Table 2). Moreover, Etrusco plants under the low Se treatment (10 µM Se) had an unexpectedly lower Se accumulation in their roots than plants grown without adding Se (Table 2).

### 2.3. Expression of Genes Coding for High-Affinity Sulfate Transporters

The expression of the two high-affinity sulfate transporters *TdSultr1.1* and *TdSultr1.3* was detected and recorded separately in the shoots and roots of each genotype grown in the low S medium (L) with the addition of Se in a high concentration (50 µM) (Figure 3).

The two high-affinity sulfate transporters showed very different expression patterns. The gene coding the transporter *TdSultr1.1* was upregulated both in the shoot and root in the genotypes Etrusco and Kamut compared to Svevo (control), whereas no significant differences were found in Tur-21. The upregulation was higher in the shoot (up to 5.85-fold in the genotype Kamut) compared to the root (up to 4.8-fold in Kamut). On the other hand, the abundance of the transcript for the gene *TdSultr1.3* was drastically reduced in all three turanicum genotypes both in the shoot and root, compared to Svevo.

### 2.4. Principal Component Analysis (PCA) to Reveal Alterations within the Ionome of Plant Shoots

To provide a more comprehensive analysis of the ionomic changes that occur when plants cope with low S stress and Se supply, PCA was performed for each different condition, and the score plots of the shoots of wheat plants are reported in Figure 4 (scatter plots and relative loadings plot of both PC1 and PC2 are reported in Appendix A).

The results of the PCA revealed two main components with over 95% variability: in the S0 condition, the first component (PC1) explained 84% of the total variance, and the second component (PC2) explained 8% of the total variance (Figure 4A). In L0, the first component (PC1) explained 76% of the total variance, and the second component (PC2) explained 22% of the total variance (Figure 4B). In S10, the first component (PC1) explained 67% of the total variance, and the second component (PC2) explained 27% of the total variance (Figure 4C). In L10, the first component (PC1) explained 76% of the total variance, and the second component (PC2) explained 13% of the total variance (Figure 4D). In S50, the first component (PC1) explained 56% of the total variance, and the second component (PC2) explained 29% of the total variance (Figure 4E). Finally, in L50, the first component (PC1) explained 77% of the total variance, and the second component (PC2) explained 19% of the total variance (Figure 4F).

In each condition, the PCA clearly separated Svevo, Tur-21, Kamut, and Etrusco into four clusters, indicating clear differences among the ionomes of the different genotypes grown under similar external conditions (Figure 4).

The Svevo shoots from both the sufficient (S) and low S medium (L) were always located on the positive axis of PC1, irrespective of Se presence and dosage, whereas the three turanicum genotypes were closer and were mostly located on the negative axis of PC1, except for Etrusco shoots from the L0 condition positioned with Svevo in the positive axis of PC1 (Figure 4B).

In the loading plot, we observed that shoot Fe, Mn, and Zn under all the different treatments contributed positively toward PC1. On the other hand, on the PC2, the positive direction was highly and mainly loaded with Fe and S, except for the L10, in which Fe and S were associated negatively towards PC2 (Figure 4).

Given the importance of Fe in PCA loading, its accumulation pattern in the shoot and root of wheat plants as a function of S nutrition and Se supply is reported in Figure 5. We found that the low S condition markedly reduced Fe accumulation in both the shoots and roots of nearly all genotypes (with reductions ranging from 40 to 50%), except for Etrusco shoots, in which S nutrition did not significantly affect the plant’s capability to accumulate Fe (Figure 5D). It is interesting that Fe accumulation also changed in response to Se supply. In particular, in Svevo and Kamut shoots grown in the S condition, the lowest Se treatment (10 µM Se) significantly increased the Fe concentration by 50 and 60%, respectively (Figure 5A,C), whereas in Tur-21 and Etrusco shoots grown in the S condition, the high Se treatment (50 µM Se) significantly reduced Fe concentration by 60 and 54%, respectively (Figure 5B,D). Additionally, at the root level, we found that the low Se treatment (10 µM Se) significantly increased Fe accumulation in Kamut roots in both the S and L conditions (by 60% and 2-fold, respectively) (Figure 5C), although under the same treatment, Fe accumulation decreased in Etrusco roots (by 25 and 45%, respectively) (Figure 5D). However, in this latter case, the high Se treatment (50 µM Se) in the S condition significantly increased root Fe concentration by 50% (Figure 5D). On the other hand, the Se supply did not significantly affect the Fe amount in the roots of Svevo (Figure 5A), whereas it significantly increased the Fe accumulation in Tur-21 roots from the S condition (by 70 and 45%, at low and high dosage, respectively) (Figure 5B).

## 3. Discussion

During the last few years, research to obtain fortified crops has been constantly increasing. Considerable work has been focused on the biofortification of cereals, in particular with Zn and Fe, leading to the development of cereal cultivars whose grains are characterized by increased bioavailable concentrations of these essential elements [21,22]. This approach ensures an adequate intake of essential micronutrients since the diet of a large part of the world’s population is limited to the use of starch-rich cereals.

Recently, attempts concerning the fortification of staple crops with Se have been increasing [23,24,25,26,27] and greater attention has been given to this element and the mechanisms regulating its homeostasis in plants ([28] and references therein). Selenium is an essential element for humans and the main source of this microelement is represented by vegetable-derived foods.

It is well known that the availability and uptake of nutrients by plants are affected by many factors in the soil–plant environment, with the interaction between nutrients being particularly important [29]. Indeed, the role of S on Se uptake by plants is well known [8,28,30], with sulfate availability in the rhizosphere being generally associated with lower selenate uptake, whereas S deficiency conditions lead to the higher uptake of Se [11].

Therefore, here, we exploited the synergistic Se/S interaction to enhance Se accumulation in durum wheat plants. In particular, we tested the hypothesis that the modulation of S availability may improve a plant’s capability to take up Se, leading to a higher Se accumulation in the plants, and this hypothesis was tested with the screening of the Se uptake ability of four tetraploid wheat genotypes, including a durum wheat cultivated genotype, cv. Svevo (*Triticum turgidum* ssp. *durum*), and three ancient Khorasan genotypes, namely Kamut, Turanicum 21, and Etrusco (*Triticum turgidum* ssp. *turanicum*).

The plants did not show significant symptoms of damage during the experimental period, but the different treatments affected both fresh biomass production and leaf greenness (Figure 1 and Figure 2). Changes in fresh biomass production were investigated in the four wheat genotypes grown under different nutritional conditions and it was found that Kamut plants showed better growth and biomass production under all conditions compared to the other genotypes.

It is known that most crops are sensitive to Se at high concentrations, whereas Se at a low dosage has been shown to stimulate plant growth [31,32,33]. Accordingly, we found that the addition of the highest Se dose (50 µM) to the low S medium slightly affected shoot development in all genotypes (in Svevo, the lowest Se concentration also reduced the shoot biomass). In roots, 50 µM Se had significant impacts on root growth, with the inhibition ranging from 10% in Tur-21 to 50% in Etrusco (Figure 1). However, the phytotoxic effect of high Se concentration was reduced by feeding the plants with an adequate supply of sulfate, as confirmed by the greatest shoot and root biomass accumulation of Kamut plants grown under the S50 condition (Figure 1).

Comparisons of leaf greenness in terms of the SPAD index showed that a low S availability without Se (L0) slightly (less than 15%) but significantly reduced chlorophyll content in all genotypes. The leaf chlorophyll amount could be limited by a reduced production of S metabolites, such as sulfide, which has been shown to prevent autophagy and senescence in Arabidopsis [34]. On the other hand, the effect of limited S availability on the uptake and accumulation of Fe [35], which plays an important role in chlorophyll biosynthesis, cannot be ruled out.

It is important to note that in contrast to previous studies in which plant exposure to Se prevents chlorophyll loss allowing an efficient net photosynthetic rate [36], in the present experiment, Se addition was accomplished with a lower chlorophyll content in both the low S and sufficient S conditions, but again the reduction was lower than 15% (Figure 2). However, our results are consistent with those reported in the literature [37], indicating that responses to Se vary depending on dose, application method, and plant species considered.

Statistical analysis showed a significant interaction between S and Se accumulation at different concentrations of Se for all genotypes (Table 1 and Table 2). As expected, S deficiency reduced total S accumulation, but Se application strongly enhanced S accumulation in all wheat lines, at least at the lowest supply of Se, demonstrating a synergic interaction between Se and S. Overall, Etrusco was characterized by an impressive ability to accumulate S: when the S-deficient plants were supplied with the highest Se supply, the roots and shoots reached a 23-fold and 7-fold higher S level than the Svevo control (Table 2).

Se accumulation depended on the level of S and Se in the solution (Table 1 and Table 2). In particular, Se uptake was not affected by Se supply in S-sufficient plants, with a few exceptions (observed for Svevo and Tur-21), according to studies reporting that the application of S inhibits Se uptake [38]. However, low S supply (L) clearly stimulated Se accumulation in all genotypes, as previously reported in *Triticum aestivum* plants showing an increase in the uptake of Se when exposed to sulfur limitation [39].

The main goal of this study was to enhance the uptake and transport of Se to the aerial parts of plants, which is a major goal in bio-fortification, to ultimately improve food security. Our data showed that S availability is an important tool to achieve this goal since a low S condition triggered Se accumulation in wheat shoots. In particular, among the three L conditions, the highest Se one (L50) resulted in the highest levels of Se in both the roots and shoots of wheat plants (Table 2).

To gain insight into the regulatory mechanisms underlying Se uptake, the relative expression patterns of two selected high-affinity sulfate transporters (*TdSultr1.1* and *TdSultr1.3*) were investigated in the shoot and root tissues of wheat lines via quantitative real-time RT-PCR.

Transmembrane proton sulfate co-transporters constitute a large family of proteins in plants [40] and their expression levels are regulated in response to changes in sulfate availability [41]. It is well known that the reduction of S supply in the growth medium results in an increase in the root uptake capacity of sulfate [40,41]. Additionally, selenate uptake and transport inside the plant are mediated by sulfate transporters [42]. It has been suggested that the rapid and significant rise of the expression of the genes encoding high-affinity sulfate transporters could be responsible for the enhanced Se influx in roots in S-deficient plants [28]. The finding that the S deficiency induced greater expression of Sultr1.1 in all tissues of *T. aestivum*, resulting in an increased accumulation of Se [11], confirms this hypothesis unambiguously.

It has been demonstrated in durum wheat that the most responsive gene to S deficiency was the high-affinity sulfate transporter *TdSultr1.1*, in both root and shoot tissues, indicating an increased sulfate uptake rate and distribution within the plant [43]. On the other hand, *TdSultr1.3*, mainly expressed in shoots, was only weakly induced upon S deprivation but was greatly upregulated in response to Fe deficiency, suggesting that some functions of *TdSultr1.1* and *TdSultr1.3* differed during evolution [43].

Additionally, in this experiment, it was proven that the expression profile of *TdSultr1.1* differed markedly from that of *TdSultr1.3* in the L50 condition. In particular, the expression of the gene coding the transporter *TdSultr1.1* was significantly higher both in the shoots and roots in Etrusco and Kamut genotypes compared to Svevo, whereas the expression of *TdSultr1.3* was drastically reduced in the three turanicum genotypes, both in the shoots and roots compared to Svevo (Figure 3). These substantial differences between the mechanisms involved in the plant uptake and distribution of sulfate and selenate in different wheat genotypes explain the different accumulation patterns detected for both elements and suggest there are many species- and genotype-specific differences regulating sulfate uptake rate. For example, in Arabidopsis plants, it has been demonstrated that SULTR1;2 is the preferred transporter for the uptake of selenate into the plant root [42]. However, in our experiment, *TdSultr1.1* could be responsible for the observed highest increase in Se accumulation in Etrusco plants, due to the withdrawal of S. Incidentally, for selenate-treated plants, *TdSultr1.3* was the most responsive gene to low S availability in Svevo, contrary to what was previously found in only S-deficient Svevo plants [43].

Based on our results, we wondered whether the interaction between S and Se homeostasis might also drive the shoot and root ionome. It is well known that nutrients interact with one another, altering their reciprocal requirements and influencing ion homeostasis [29] For example, the coordination of S and Fe homeostasis shows precisely how a deficiency in one element (herein S) leads to a lower accumulation of the other ion (herein Fe), and conversely, how a deficiency in one element (herein Fe) leads to a greater demand for the other ion (herein S) [35].

We then examined the nutrient status of shoots by measuring the levels of both macro- and micronutrients in these four wheat lines. As expected, the different nutritional conditions modified the ionomic composition of plant tissues.

In each condition, the PCA clearly separated the four different genotypes, indicating clear differences between the ionomes of the different genotypes grown under similar external conditions (Figure 4). Irrespective of S and Se concentration, Svevo was always located on the positive axis of PC1, with Fe, Mn, and Zn contributing positively towards PC1, whereas the three turanicum genotypes were closer and were mostly located on the negative axis of PC1. These differences might be consistent with a different ability of these wheat lines to accumulate nutrients [44].

According to previous studies [35], the low S condition markedly reduced Fe accumulation in both the shoots and roots of most genotypes (with reductions ranging from 40 to 50%), except for Etrusco shoots, in which S nutrition did not significantly affect the plant’s capability to accumulate Fe (Figure 5).

More interestingly, a beneficial effect of Se supply was found, at least at a low dosage, on Fe accumulation in plant tissues (Figure 5). In particular, low Se treatment (10 µM Se) significantly increased Fe concentration in Svevo and Kamut shoots grown in the optimum S condition, by 50 and 60%, respectively. On the other hand, high Se treatment (50 µM Se) significantly reduced Fe concentration in Tur-21 and Etrusco shoots grown in the S condition, by 60 and 54%, respectively.

Even if the mechanisms driving Fe transport and accumulation in a Se-dependent manner are yet to be identified, this finding offers a great opportunity to go beyond using Se only to enhance its content in plant tissues.

## 4. Materials and Methods

### 4.1. Plant Growth

Seeds of *Triticum turgidum* (cv. Svevo, Tur-21, Kamut, and Etrusco) were sown on moistened paper and germinated in the dark at 20 °C for 4 days. The seedlings were then transferred to a growth chamber in a hydroponics facility. The growth conditions were 27/20 °C with 14/10 h day/night cycles and a relative humidity of 80% and 200 mmol m^−2^ s^−1^ PAR at leaf level. Briefly, sets of six seedlings were transplanted to a container filled with 2.2 L of nutrient solution (NS) [45], while being exposed to two different sulfate levels (S and L, 1.2 and 0.06 mM sulfate, respectively) and three different Se levels (0, 10, and 50 μM), provided as selenate. After 20 days, the plants were harvested and separated into roots and shoots.

### 4.2. Chlorophyll Content

The concentration of chlorophyll content per unit area was estimated in attached leaves using a SPAD portable apparatus (Minolta Co., Osaka, Japan), using the first fully expanded leaf from the top of the plant.

### 4.3. Analysis of Micro- and Macronutrient Concentrations

A total of 12 elements (Ca, Cu, Fe, K, Mg, Mn, Mo, Na, P, S, Se, and Zn) were quantified using ICP-OES (Spectro Arcos, Spectro Ametek, Kleve, Germany), according to Celletti et al. (2016) [46]. Briefly, shoot and root tissues were oven-dried at 60 °C to a constant weight. Dry tissues were finely ground and then weighed. Subsequently, the tissues were digested with concentrated ultrapure HNO_3_ (65% *v*/*v*, Carlo Erba, Milano, Italy) using a single reaction chamber microwave digestion system (UltraWAVE, Milestone, Shelton, CT, USA). The element concentrations were analyzed using ICP-OES, and using spinach leaves (SRM 1570a) and tomato leaves (SRM 1573a) as external certified reference materials.

### 4.4. Total RNA Extraction and RT-PCR Analysis

Total RNA was extracted from the shoots and roots of hydroponically grown plants using a Spectrum Plant Total RNA kit (Sigma-Aldrich, St. Louis, MO, USA). Single-strand cDNA was achieved starting from 1 µg of total RNA using a QuantiTect Reverse Transcription Kit (Qiagen, Hilden, Germany). qRT-PCR was performed using a CFX 96 Real-Time PCR Detection System device (Bio-Rad, Hercules, CA, USA), following the procedure described by Camerlengo et al. (2017) [47]. Each reaction was performed in a final volume of 15 µL, made of 7.5 µL SsoAdvUniver SYBR GRN SMX (Bio-Rad), 0.5 µM of each primer, and 1 µL of cDNA. The protocol of amplification was described in Sestili et al. (2019) [48] and consisted of 94 °C for 30 s and 40 cycles at 94 °C for 5 s, 60 °C for 30 s, and a melt curve of 65–95 °C with a 0.5 °C increment at 5 s/step. β-actin was used as the housekeeping gene. Relative gene expressions were calculated using the 2^−∆∆Ct^ method [49]. The specific primer pairs were designed within the 3′ end region of the two genes coding for Group 1 high-affinity sulfate transporters in durum wheat (*TdST1.1* and *TdST1.3*), based on cDNA sequences previously isolated by Ciaffi et al. [43]. Data were achieved from three biological replicates for genotype, each of which, in turn, was associated with three technical replicates.

### 4.5. Statistical Analysis

Each reported value represents the mean ± SD of measurements carried out in triplicate (n = 3) and obtained from three independent experiments. The significance between treatment means was compared by employing a one-way analysis of variance (ANOVA), using the statistical software CoStat Version 6.45 (CoHort, Berkeley, CA, USA). When the effect of treatments was significant, the means were separated by the least significant difference (LSD) test at *p* < 0.05.

The effects of S availability treatment, genotype, and interaction between them were evaluated at two levels of significance: *p* < 0.01 (**) and *p* < 0.001 (***). Data were subjected to two-way ANOVA and the Fisher’s test was used for mean separation and to provide homogeneous groups for the means (at *p* < 0.05). All data were processed in XLSTAT v. 2022.4.5, a user-friendly statistical software for Microsoft Excel.

## 5. Conclusions

The four wheat genotypes exhibited considerable genotypic variation and capacity to accumulate S and Se in response to their available supply. Etrusco was characterized by an impressive ability to accumulate both S and Se.

The two high-affinity sulfate transporters (*TdSultr1.1* and *TdSultr1.3*) showed different expression patterns, suggesting that the mechanism of sulfate uptake regulation under S deficiency and Se supply might be different in different wheat genotypes.

Interactions between nutrients impact plant nutrient status: low S supply (L) clearly stimulated Se accumulation in wheat genotypes while reducing Fe accumulation in almost all genotypes.

Fe content in wheat genotypes was generally enhanced by Se treatment, suggesting a synergistic effect of selenate treatment on Fe accumulation.

These findings highlight the importance of exploring wheat biodiversity, offering a significant insight into the sustainable use of S nutrition for crop Se biofortification, with crucial significance in improving human nutrition and health.

## Figures and Tables

**Figure 1 ijms-24-05443-f001:**
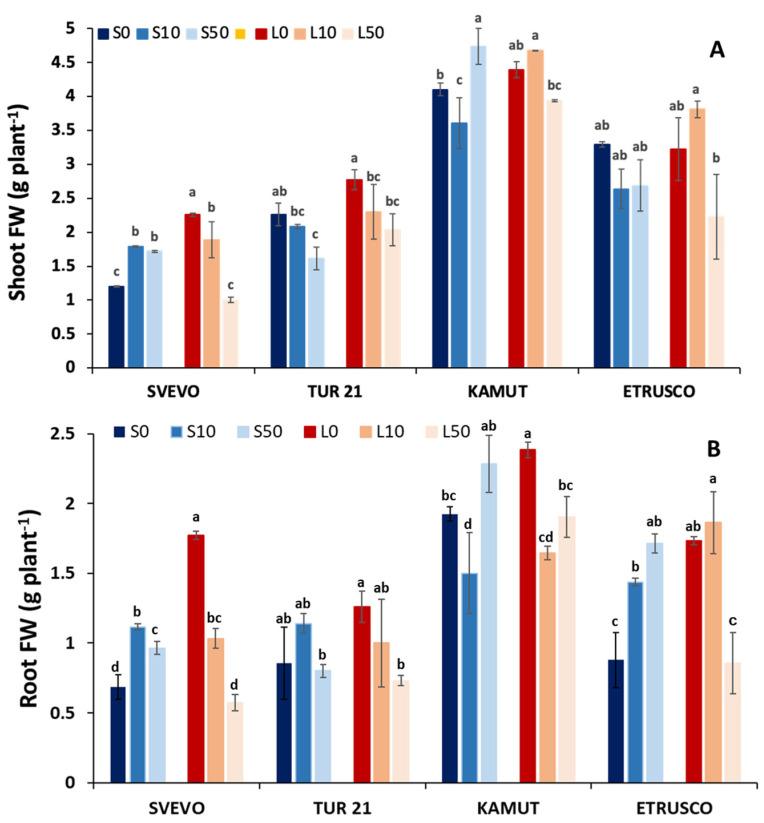
Plant growth parameters. Shoot (**A**) and root (**B**) fresh weight of four wheat genotypes (Svevo, Tur-21, Kamut, and Etrusco). Plants were grown while exposed to different sulfate levels (S and L, 1.2 and 0.06 mM sulfate, respectively) and three different Se levels (0, 10, and 50 µM), provided as selenate. Data are means ± SD of four independent replications run in triplicate. Significant differences between samples are indicated by different letters (*p* < 0.05).

**Figure 2 ijms-24-05443-f002:**
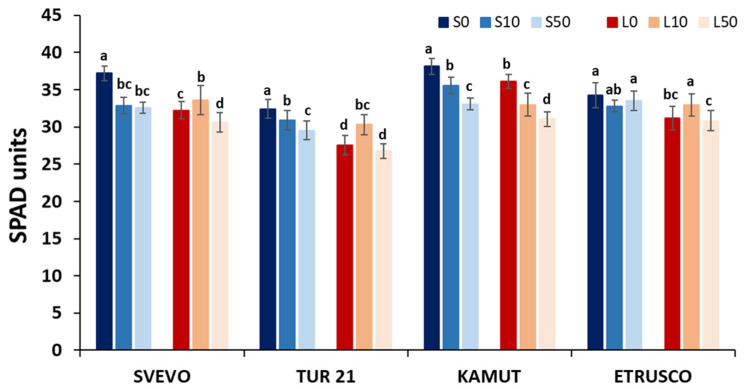
Chlorophyll content. Chlorophyll concentrations were measured using a SPAD meter in the leaves of four wheat genotypes (Svevo, Tur-21, Kamut, and Etrusco). Plants were grown while exposed to different sulfate levels (S and L, 1.2 and 0.06 mM sulfate, respectively) and three different Se levels (0, 10, and 50 mM), provided as selenate. SPAD readings were made using the first fully expanded leaf from the top of the plant. Significant differences between samples are indicated by different letters (*p* < 0.05).

**Figure 3 ijms-24-05443-f003:**
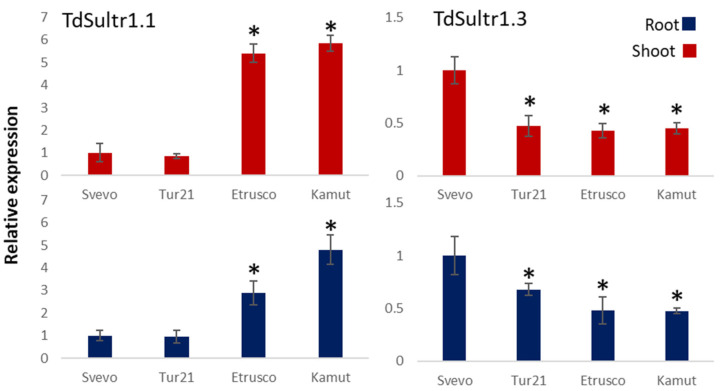
Expression of genes coding for high-affinity sulfate transporters. Expression of genes coding for two high-affinity sulfate transporters (TdST1.1 and TdST1.3) in root and shoot tissues of four wheat genotypes (Svevo, Tur-21, Kamut, and Etrusco) by qRT-PCR. Plants were grown being exposed to low sulfate level (L, 0.06 mM sulfate) and 50 mM Se, provided as selenate. Data are shown as fold differences in transcript abundance between the control (cv. Svevo) and three tetraploid genotypes belonging to ssp. turanicum (Tur21, Etrusco, and Kamut). Standard error bars are indicated. Genes significantly (*p* < 0.05) up- or downregulated are marked by an asterisk (*).

**Figure 4 ijms-24-05443-f004:**
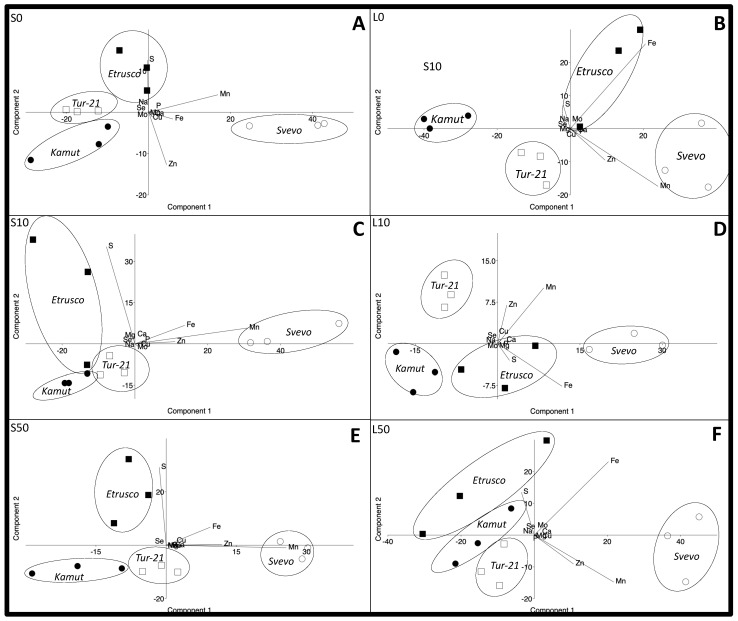
Principal component analysis (PCA) to reveal changes within the ionome of plant shoots. Principal component analysis (PCA) scatter plot of the shoot ionome of four wheat genotypes (Svevo, Tur-21, Kamut, and Etrusco). Plants were grown while exposed to different sulfate levels (S and L, 1.2 and 0.06 mM sulfate, respectively) and three different Se levels (0, 10, and 50 mM), provided as selenate. (**A**): S0 condition, (**B**): L0 condition, (**C**): S10 condition, (**D**): L10 condition, (**E**): S50 condition, (**F**): L50 condition.

**Figure 5 ijms-24-05443-f005:**
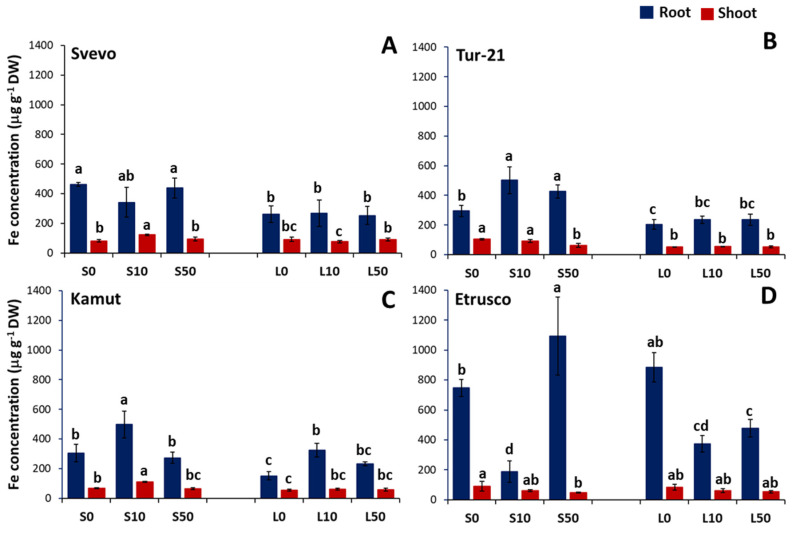
Plant capability to accumulate Fe as a function of S availability and Se treatment. Iron accumulation in roots and shoots of four wheat genotypes: Svevo (**A**), Tur-21 (**B**), Kamut (**C**), and Etrusco (**D**). Plants were grown while exposed to different sulfate levels (S and L, 1.2 and 0.06 mM sulfate, respectively) and three different Se levels (0, 10, and 50 μM), provided as selenate. Significant differences between samples are indicated by different letters (*p* < 0.05).

**Table 1 ijms-24-05443-t001:** Root and shoot sulfur and selenium concentrations of four wheat genotypes (Svevo, Tur-21, Kamut, and Etrusco) grown with sufficient (S) availability of sulfate.

Genotype	Treatment	S (mg g^−1^ DW)		Se (μg g^−1^ DW)	
		Root	Shoot	Root	Shoot
**Svevo**	S0	2.09 ± 0.09 c	3.20 ± 0.15 c	0 ± 0 d	0.40 ± 0 bc
	S10	2.12 ± 0.04 c	7.34 ± 1.55 c	0 ± 0 d	0.64 ± 0.01 ab
	S50	6.0 ± 0.88 c	9.60 ± 0.45 c	1.15 ± 0.01 cd	0.69 ± 0.09 a
**Tur-21**	S0	1.73 ± 0.01 c	2.79 ± 0.28 c	1.79 ± 0.37 cd	0.65 ± 0.12 a
	S10	8.39 ± 0.41 c	5.49 ± 0.17 c	3.09 ± 0.47 bc	0.55 ± 0.11 abc
	S50	29.16 ± 1.53 bc	5.39 ± 1.24 c	1.93 ± 0.04 cd	0.64 ± 0.07 ab
**Kamut**	S0	8.79 ± 1.05 c	2.76 ± 0.05 c	1.70 ± 0.32 cd	0.50 ± 0.07 abc
	S10	21.47 ± 9.88 bc	4.18 ± 0.41 c	2.46 ± 0.45 c	0.47 ± 0.07 abc
	S50	3.18 ± 0.51 c	2.93 ± 0.07 c	2.32 ± 0.03 c	0.59 ± 0.08 abc
**Etrusco**	S0	11.46 ± 3.61 c	18.45 ± 1.34 b	6.33 ± 2.05 a	0.36 ± 0.01 c
	S10	48.99 ± 28.82 b	49.24 ± 6.91 a	3.22 ± 1.24 bc	0.67 ± 0.12 a
	S50	100.97 ± 12.45 a	25.31 ± 5.38 b	4.97 ± 0.29 ab	0.66 ± 0.08 a
**Source of variation**					
**Genotype**		1.62 × 10^−10^ ***	1.074 × 10^−17^ ***	5.79779 × 10^−11^ ***	0.146132 -
**Treatment**		6.91 × 10^−7^ ***	1.541 × 10^−8^ ***	0.416611378 -	0.000155 ***
**Genotype*Treatment**		2.2 × 10^−7^ ***	2.417 × 10^−9^ ***	0.000473439 ***	0.002185 **

Data are represented as mean ± SD. Different letters within the same columns indicating significant differences according to Fisher’s test (*p* < 0.05). Significant effect: ** *p* < 0.01; *** *p* < 0.001; - not significant.

**Table 2 ijms-24-05443-t002:** Root and shoot sulfur and selenium concentrations of four wheat genotypes (Svevo, Tur-21, Kamut, and Etrusco) grown under limited (L) availability of sulfate.

Genotype	Treatment	S (mg g^−1^ DW)		Se (μg g^−1^ DW)	
		Root	Shoot	Root	Shoot
**Svevo**	L0	1.61 ± 0.01 c	2.76 ± 0.09 f	0 ± 0 d	0 ± 0 d
	L10	6.37 ± 1.20 c	10.21 ± 0.34 cd	1.77± 0.08 c	0.98 ± 0.12 c
	L50	1.76 ± 0.23 c	3.22 ± 0.66 f	3.036 ± 0.10 bc	4.09 ± 0.18 a
**Tur-21**	L0	1.73 ± 0.04 c	5.61 ± 0.89 ef	2.55 ± 0.04 c	0.81 ± 0.046 c
	L10	10.09 ± 1.46 c	3.63 ± 1.03 ef	2.95 ± 0.16 bc	1.03 ± 0.13 c
	L50	10.01 ± 2.78 c	7.41 ± 0.57 de	2.56 ± 0.69 c	3.17 ± 0.13 b
**Kamut**	L0	2.19 ± 0.03 c	11.75 ± 3.06 c	1.34 ± 0.02 cd	0.58 ± 0.01 c
	L10	3.29 ± 1.32 c	2.74 ± 0.42 f	2.59 ± 0.23 c	1.10 ± 0.16 c
	L50	5.25 ± 0.53 c	3.55 ± 0.25 ef	4.48 ± 0.39 b	3.45 ± 0.18 b
**Etrusco**	L0	41.11 ± 3.09 b	11.93 ± 2.18 c	7.064 ± 1.68 a	1.35 ± 0.29 c
	L10	76.26 ± 20.64 a	18.40 ± 0.99 b	2.50 ± 0.64 c	1.15 ± 0.18 c
	L50	98.76 ± 29.23 a	23.45 ± 2.10 a	6.99 ± 0.30 a	3.68 ± 0.43ab
**Source of variation**					
**Genotype**		2.06 × 10^−13^ ***	1.3555 × 10^−16^ ***	3.08 × 10^−12^ ***	0.002313 **
**Treatment**		0.001401 **	0.06291953 -	1.11 × 10^−7^ ***	1.44 × 10^−21^ ***
**Genotype*Treatment**		0.00158 **	1.83 × 10^−11^ ***	6.97 × 10^−9^ ***	3.04 × 10^−6^ ***

Data are represented as mean ± SD. Different letters within the same columns indicate significant differences according to Fisher’s test (*p* < 0.05). Significant effect: ** *p* < 0.01; *** *p* < 0.001; - not significant.

## Data Availability

Not applicable.

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
