# Peer review of "Interaction between Sulfate and Selenate in Tetraploid Wheat (*Triticum turgidum* L.) Genotypes"

_ijms, 2023, doi:10.3390/ijms24065443_

Round 1

Reviewer 1 Report

Good efforts but minor corrections are needed!

Some comments:

1. L51: please change SeO42- in subscript instead of SeO42-.

2. L89-96: Rewrite that paragraph to highlight the purpose of the study clearly.

3. L97-104 please remove that part it’s repeated information you mentioned it in different places.

4. Grammar check is needed and some words have typo mistakes.

5. Results section: the explanation of the results is confusing especially when the authors use a different synonymous for exp. Sulfate limited supply and a low-S stimulatory, Low S availability, and low-S medium, nutritional treatments! in my opinion, it’s better to use the same expression to deliver the meaning easily to the reader and make it more clear.

6. L122: this sentence is not clear! Does it result in S50 or L50?

7. L158: what do you mean by “normal-S condition”?

8. L449: remove the link of the growth chamber.

Other comments:

·       L48: I think the oxidation states don’t include 0 as it’s known as an atomic state, not an oxidant so please check that!

·       L123: What do you mean by “control plants”  and L139: “ the control’ is it without any treatment or do you mean S0 and L0?

Author Response

The authors would like to thank the reviewers for their constructive comments and useful suggestions for improving our manuscript and have taken all of them into account in producing the attached revised version. We feel that their input greatly helped to improve this manuscript.

We believe that we have addressed all the reviewers’ comments by providing clarification of the writing as well as by streamlining the manuscript to provide a clear reflection of our work in the current version. A detailed point-by-point in annotated summary follows below. We fully hope that the manuscript is now of suitable quality for publication.

Reviewer 1

Good efforts but minor corrections are needed!

We are very happy to receive this comment and we truly appreciate your support of our work.

Some comments:

  1. L51: please change SeO42in subscript instead of SeO42-.

Done

  1. L89-96: Rewrite that paragraph to highlight the purpose of the study clearly.

According to the reviewer’s comment, the text has been modified (l.87).

  1. L97-104 please remove that part it’s repeated information you mentioned it in different places.

Done

  1. Grammar check is needed and some words have typo mistakes.

Done

  1. Results section: the explanation of the results is confusing especially when the authors use a different synonymous for exp. Sulfate limited supply and a low-S stimulatory, Low S availability, and low-S medium, nutritional treatments! in my opinion, it’s better to use the same expression to deliver the meaning easily to the reader and make it more clear.

According to the reviewer’s comment, we have tried to standardize the terminology as much as possible.

  1. L122: this sentence is not clear! Does it result in S50 or L50?

This paragraph (l.112) refers to the S0 condition, the next one (l.119) is related to the L0 condition.

  1. L158: what do you mean by “normal-S condition”?

According to the reviewer’s comment, “normal” was replaced by “sufficient”.

  1. L449: remove the link of the growth chamber.

Done

Other comments:

  • L48: I think the oxidation states don’t include 0 as it’s known as an atomic state, not an oxidant so please check that!

We checked it and it was reported in the same way in other papers (as well as in cited reference [8].

  • L123: What do you mean by “control plants”  and L139: “ the control’ is it without a

    The authors would like to thank the reviewers for their constructive comments and useful suggestions for improving our manuscript and have taken all of them into account in producing the attached revised version. We feel that their input greatly helped to improve this manuscript.

    We believe that we have addressed all the reviewers’ comments by providing clarification of the writing as well as by streamlining the manuscript to provide a clear reflection of our work in the current version. A detailed point-by-point in annotated summary follows below. We fully hope that the manuscript is now of suitable quality for publication.

    Reviewer 1

    Good efforts but minor corrections are needed!

    We are very happy to receive this comment and we truly appreciate your support of our work.

    Some comments:

    1. L51: please change SeO42in subscript instead of SeO42-.

    Done

    1. L89-96: Rewrite that paragraph to highlight the purpose of the study clearly.

    According to the reviewer’s comment, the text has been modified (l.87).

    1. L97-104 please remove that part it’s repeated information you mentioned it in different places.

    Done

    1. Grammar check is needed and some words have typo mistakes.

    Done

    1. Results section: the explanation of the results is confusing especially when the authors use a different synonymous for exp. Sulfate limited supply and a low-S stimulatory, Low S availability, and low-S medium, nutritional treatments! in my opinion, it’s better to use the same expression to deliver the meaning easily to the reader and make it more clear.

    According to the reviewer’s comment, we have tried to standardize the terminology as much as possible.

    1. L122: this sentence is not clear! Does it result in S50 or L50?

    This paragraph (l.112) refers to the S0 condition, the next one (l.119) is related to the L0 condition.

    1. L158: what do you mean by “normal-S condition”?

    According to the reviewer’s comment, “normal” was replaced by “sufficient”.

    1. L449: remove the link of the growth chamber.

    Done

    Other comments:

    • L48: I think the oxidation states don’t include 0 as it’s known as an atomic state, not an oxidant so please check that!

    We checked it and it was reported in the same way in other papers (as well as in cited reference [8].

    • L123: What do you mean by “control plants”  and L139: “ the control’ is it without any treatment or do you mean S0 and L0?

    In the first case “control” refers to the same S condition (S) but without Se, in the second one “control” refers to the same S condition (L) and again without Se.

    The authors would like to thank the reviewers for their constructive comments and useful suggestions for improving our manuscript and have taken all of them into account in producing the attached revised version. We feel that their input greatly helped to improve this manuscript.

    We believe that we have addressed all the reviewers’ comments by providing clarification of the writing as well as by streamlining the manuscript to provide a clear reflection of our work in the current version. A detailed point-by-point in annotated summary follows below. We fully hope that the manuscript is now of suitable quality for publication.

    Reviewer 1

    Good efforts but minor corrections are needed!

    We are very happy to receive this comment and we truly appreciate your support of our work.

    Some comments:

    1. L51: please change SeO42in subscript instead of SeO42-.

    Done

    1. L89-96: Rewrite that paragraph to highlight the purpose of the study clearly.

    According to the reviewer’s comment, the text has been modified (l.87).

    1. L97-104 please remove that part it’s repeated information you mentioned it in different places.

    Done

    1. Grammar check is needed and some words have typo mistakes.

    Done

    1. Results section: the explanation of the results is confusing especially when the authors use a different synonymous for exp. Sulfate limited supply and a low-S stimulatory, Low S availability, and low-S medium, nutritional treatments! in my opinion, it’s better to use the same expression to deliver the meaning easily to the reader and make it more clear.

    According to the reviewer’s comment, we have tried to standardize the terminology as much as possible.

    1. L122: this sentence is not clear! Does it result in S50 or L50?

    This paragraph (l.112) refers to the S0 condition, the next one (l.119) is related to the L0 condition.

    1. L158: what do you mean by “normal-S condition”?

    According to the reviewer’s comment, “normal” was replaced by “sufficient”.

    1. L449: remove the link of the growth chamber.

    Done

    Other comments:

    • L48: I think the oxidation states don’t include 0 as it’s known as an atomic state, not an oxidant so please check that!

    We checked it and it was reported in the same way in other papers (as well as in cited reference [8].

    • L123: What do you mean by “control plants”  and L139: “ the control’ is it without any treatment or do you mean S0 and L0?

    In the first case “control” refers to the same S condition (S) but without Se, in the second one “control” refers to the same S condition (L) and again without Se.

    The authors would like to thank the reviewers for their constructive comments and useful suggestions for improving our manuscript and have taken all of them into account in producing the attached revised version. We feel that their input greatly helped to improve this manuscript.

    We believe that we have addressed all the reviewers’ comments by providing clarification of the writing as well as by streamlining the manuscript to provide a clear reflection of our work in the current version. A detailed point-by-point in annotated summary follows below. We fully hope that the manuscript is now of suitable quality for publication.

    Reviewer 1

    Good efforts but minor corrections are needed!

    We are very happy to receive this comment and we truly appreciate your support of our work.

    Some comments:

    1. L51: please change SeO42in subscript instead of SeO42-.

    Done

    1. L89-96: Rewrite that paragraph to highlight the purpose of the study clearly.

    According to the reviewer’s comment, the text has been modified (l.87).

    1. L97-104 please remove that part it’s repeated information you mentioned it in different places.

    Done

    1. Grammar check is needed and some words have typo mistakes.

    Done

    1. Results section: the explanation of the results is confusing especially when the authors use a different synonymous for exp. Sulfate limited supply and a low-S stimulatory, Low S availability, and low-S medium, nutritional treatments! in my opinion, it’s better to use the same expression to deliver the meaning easily to the reader and make it more clear.

    According to the reviewer’s comment, we have tried to standardize the terminology as much as possible.

    1. L122: this sentence is not clear! Does it result in S50 or L50?

    This paragraph (l.112) refers to the S0 condition, the next one (l.119) is related to the L0 condition.

    1. L158: what do you mean by “normal-S condition”?

    According to the reviewer’s comment, “normal” was replaced by “sufficient”.

    1. L449: remove the link of the growth chamber.

    Done

    Other comments:

    • L48: I think the oxidation states don’t include 0 as it’s known as an atomic state, not an oxidant so please check that!

    We checked it and it was reported in the same way in other papers (as well as in cited reference [8].

    • L123: What do you mean by “control plants”  and L139: “ the control’ is it without any treatment or do you mean S0 and L0?

    In the first case “control” refers to the same S condition (S) but without Se, in the second one “control” refers to the same S condition (L) and again without Se.

    The authors would like to thank the reviewers for their constructive comments and useful suggestions for improving our manuscript and have taken all of them into account in producing the attached revised version. We feel that their input greatly helped to improve this manuscript.

    We believe that we have addressed all the reviewers’ comments by providing clarification of the writing as well as by streamlining the manuscript to provide a clear reflection of our work in the current version. A detailed point-by-point in annotated summary follows below. We fully hope that the manuscript is now of suitable quality for publication.

    Reviewer 1

    Good efforts but minor corrections are needed!

    We are very happy to receive this comment and we truly appreciate your support of our work.

    Some comments:

    1. L51: please change SeO42in subscript instead of SeO42-.

    Done

    1. L89-96: Rewrite that paragraph to highlight the purpose of the study clearly.

    According to the reviewer’s comment, the text has been modified (l.87).

    1. L97-104 please remove that part it’s repeated information you mentioned it in different places.

    Done

    1. Grammar check is needed and some words have typo mistakes.

    Done

    1. Results section: the explanation of the results is confusing especially when the authors use a different synonymous for exp. Sulfate limited supply and a low-S stimulatory, Low S availability, and low-S medium, nutritional treatments! in my opinion, it’s better to use the same expression to deliver the meaning easily to the reader and make it more clear.

    According to the reviewer’s comment, we have tried to standardize the terminology as much as possible.

    1. L122: this sentence is not clear! Does it result in S50 or L50?

    This paragraph (l.112) refers to the S0 condition, the next one (l.119) is related to the L0 condition.

    1. L158: what do you mean by “normal-S condition”?

    According to the reviewer’s comment, “normal” was replaced by “sufficient”.

    1. L449: remove the link of the growth chamber.

    Done

    Other comments:

    • L48: I think the oxidation states don’t include 0 as it’s known as an atomic state, not an oxidant so please check that!

    We checked it and it was reported in the same way in other papers (as well as in cited reference [8].

    • L123: What do you mean by “control plants”  and L139: “ the control’ is it without any treatment or do you mean S0 and L0?

    In the first case “control” refers to the same S condition (S) but without Se, in the second one “control” refers to the same S condition (L) and again without Se.

    The authors would like to thank the reviewers for their constructive comments and useful suggestions for improving our manuscript and have taken all of them into account in producing the attached revised version. We feel that their input greatly helped to improve this manuscript.

    We believe that we have addressed all the reviewers’ comments by providing clarification of the writing as well as by streamlining the manuscript to provide a clear reflection of our work in the current version. A detailed point-by-point in annotated summary follows below. We fully hope that the manuscript is now of suitable quality for publication.

    Reviewer 1

    Good efforts but minor corrections are needed!

    We are very happy to receive this comment and we truly appreciate your support of our work.

    Some comments:

    1. L51: please change SeO42in subscript instead of SeO42-.

    Done

    1. L89-96: Rewrite that paragraph to highlight the purpose of the study clearly.

    According to the reviewer’s comment, the text has been modified (l.87).

    1. L97-104 please remove that part it’s repeated information you mentioned it in different places.

    Done

    1. Grammar check is needed and some words have typo mistakes.

    Done

    1. Results section: the explanation of the results is confusing especially when the authors use a different synonymous for exp. Sulfate limited supply and a low-S stimulatory, Low S availability, and low-S medium, nutritional treatments! in my opinion, it’s better to use the same expression to deliver the meaning easily to the reader and make it more clear.

    According to the reviewer’s comment, we have tried to standardize the terminology as much as possible.

    1. L122: this sentence is not clear! Does it result in S50 or L50?

    This paragraph (l.112) refers to the S0 condition, the next one (l.119) is related to the L0 condition.

    1. L158: what do you mean by “normal-S condition”?

    According to the reviewer’s comment, “normal” was replaced by “sufficient”.

    1. L449: remove the link of the growth chamber.

    Done

    Other comments:

    • L48: I think the oxidation states don’t include 0 as it’s known as an atomic state, not an oxidant so please check that!

    We checked it and it was reported in the same way in other papers (as well as in cited reference [8].

    • L123: What do you mean by “control plants”  and L139: “ the control’ is it without any treatment or do you mean S0 and L0?

    In the first case “control” refers to the same S condition (S) but without Se, in the second one “control” refers to the same S condition (L) and again without Se.

    The authors would like to thank the reviewers for their constructive comments and useful suggestions for improving our manuscript and have taken all of them into account in producing the attached revised version. We feel that their input greatly helped to improve this manuscript.

    We believe that we have addressed all the reviewers’ comments by providing clarification of the writing as well as by streamlining the manuscript to provide a clear reflection of our work in the current version. A detailed point-by-point in annotated summary follows below. We fully hope that the manuscript is now of suitable quality for publication.

    Reviewer 1

    Good efforts but minor corrections are needed!

    We are very happy to receive this comment and we truly appreciate your support of our work.

    Some comments:

    1. L51: please change SeO42in subscript instead of SeO42-.

    Done

    1. L89-96: Rewrite that paragraph to highlight the purpose of the study clearly.

    According to the reviewer’s comment, the text has been modified (l.87).

    1. L97-104 please remove that part it’s repeated information you mentioned it in different places.

    Done

    1. Grammar check is needed and some words have typo mistakes.

    Done

    1. Results section: the explanation of the results is confusing especially when the authors use a different synonymous for exp. Sulfate limited supply and a low-S stimulatory, Low S availability, and low-S medium, nutritional treatments! in my opinion, it’s better to use the same expression to deliver the meaning easily to the reader and make it more clear.

    According to the reviewer’s comment, we have tried to standardize the terminology as much as possible.

    1. L122: this sentence is not clear! Does it result in S50 or L50?

    This paragraph (l.112) refers to the S0 condition, the next one (l.119) is related to the L0 condition.

    1. L158: what do you mean by “normal-S condition”?

    According to the reviewer’s comment, “normal” was replaced by “sufficient”.

    1. L449: remove the link of the growth chamber.

    Done

    Other comments:

    • L48: I think the oxidation states don’t include 0 as it’s known as an atomic state, not an oxidant so please check that!

    We checked it and it was reported in the same way in other papers (as well as in cited reference [8].

    • L123: What do you mean by “control plants”  and L139: “ the control’ is it without any treatment or do you mean S0 and L0?

    In the first case “control” refers to the same S condition (S) but without Se, in the second one “control” refers to the same S condition (L) and again without Se.

    The authors would like to thank the reviewers for their constructive comments and useful suggestions for improving our manuscript and have taken all of them into account in producing the attached revised version. We feel that their input greatly helped to improve this manuscript.

    We believe that we have addressed all the reviewers’ comments by providing clarification of the writing as well as by streamlining the manuscript to provide a clear reflection of our work in the current version. A detailed point-by-point in annotated summary follows below. We fully hope that the manuscript is now of suitable quality for publication.

    Reviewer 1

    Good efforts but minor corrections are needed!

    We are very happy to receive this comment and we truly appreciate your support of our work.

    Some comments:

    1. L51: please change SeO42in subscript instead of SeO42-.

    Done

    1. L89-96: Rewrite that paragraph to highlight the purpose of the study clearly.

    According to the reviewer’s comment, the text has been modified (l.87).

    1. L97-104 please remove that part it’s repeated information you mentioned it in different places.

    Done

    1. Grammar check is needed and some words have typo mistakes.

    Done

    1. Results section: the explanation of the results is confusing especially when the authors use a different synonymous for exp. Sulfate limited supply and a low-S stimulatory, Low S availability, and low-S medium, nutritional treatments! in my opinion, it’s better to use the same expression to deliver the meaning easily to the reader and make it more clear.

    According to the reviewer’s comment, we have tried to standardize the terminology as much as possible.

    1. L122: this sentence is not clear! Does it result in S50 or L50?

    This paragraph (l.112) refers to the S0 condition, the next one (l.119) is related to the L0 condition.

    1. L158: what do you mean by “normal-S condition”?

    According to the reviewer’s comment, “normal” was replaced by “sufficient”.

    1. L449: remove the link of the growth chamber.

    Done

    Other comments:

    • L48: I think the oxidation states don’t include 0 as it’s known as an atomic state, not an oxidant so please check that!

    We checked it and it was reported in the same way in other papers (as well as in cited reference [8].

    • L123: What do you mean by “control plants”  and L139: “ the control’ is it without any treatment or do you mean S0 and L0?

    In the first case “control” refers to the same S condition (S) but without Se, in the second one “control” refers to the same S condition (L) and again without Se.

    The authors would like to thank the reviewers for their constructive comments and useful suggestions for improving our manuscript and have taken all of them into account in producing the attached revised version. We feel that their input greatly helped to improve this manuscript.

    We believe that we have addressed all the reviewers’ comments by providing clarification of the writing as well as by streamlining the manuscript to provide a clear reflection of our work in the current version. A detailed point-by-point in annotated summary follows below. We fully hope that the manuscript is now of suitable quality for publication.

    Reviewer 1

    Good efforts but minor corrections are needed!

    We are very happy to receive this comment and we truly appreciate your support of our work.

    Some comments:

    1. L51: please change SeO42in subscript instead of SeO42-.

    Done

    1. L89-96: Rewrite that paragraph to highlight the purpose of the study clearly.

    According to the reviewer’s comment, the text has been modified (l.87).

    1. L97-104 please remove that part it’s repeated information you mentioned it in different places.

    Done

    1. Grammar check is needed and some words have typo mistakes.

    Done

    1. Results section: the explanation of the results is confusing especially when the authors use a different synonymous for exp. Sulfate limited supply and a low-S stimulatory, Low S availability, and low-S medium, nutritional treatments! in my opinion, it’s better to use the same expression to deliver the meaning easily to the reader and make it more clear.

    According to the reviewer’s comment, we have tried to standardize the terminology as much as possible.

    1. L122: this sentence is not clear! Does it result in S50 or L50?

    This paragraph (l.112) refers to the S0 condition, the next one (l.119) is related to the L0 condition.

    1. L158: what do you mean by “normal-S condition”?

    According to the reviewer’s comment, “normal” was replaced by “sufficient”.

    1. L449: remove the link of the growth chamber.

    Done

    Other comments:

    • L48: I think the oxidation states don’t include 0 as it’s known as an atomic state, not an oxidant so please check that!

    We checked it and it was reported in the same way in other papers (as well as in cited reference [8].

    • L123: What do you mean by “control plants”  and L139: “ the control’ is it without any treatment or do you mean S0 and L0?

    In the first case “control” refers to the same S condition (S) but without Se, in the second one “control” refers to the same S condition (L) and again without Se.

    ny treatment or do you mean S0 and L0?

In the first case “control” refers to the same S condition (S) but without Se, in the second one “control” refers to the same S condition (L) and again without Se.

Reviewer 2 Report

This paper needs major revision before publication.

1- There is a too lengthy introductory portion at the start of Abstract. Reduce it to maximum of two sentences.

2- Write Abstract in past tense.

3- Last paragraph of Introduction is highly confusing . If it is the work of any other scientist then it should be with a reference. If it is a part of the present study then it should be deleted from Introduction. 

4- Add aims of the present study at the end of Introduction. 

5- In Figure 1 A, standard error on the last bar is too high. Please check the raw data. It seems that values in one replicate are markedly different from the values in other replicates. If so, adjust this value by taking the mean of other replicates. Then apply statistical analysis of the whole data of this figure that will be different from the present one.

6- In Figure 3, put * in the centers of the bars.

Statistical analysis in Figure 5 seems highly confusing. Apparently it seems wrong. Bars with same height have different letters and those with different heights have same height. Recheck please.

7- Format refences uniformly and correctly. Some references are incomplete. Write scientific names in Italics. Write names of journals in abbreviated form. 

8- Improve formatting throughout the paper. Especially check the chemical formulas. 

Author Response

The authors would like to thank the reviewers for their constructive comments and useful suggestions for improving our manuscript and have taken all of them into account in producing the attached revised version. We feel that their input greatly helped to improve this manuscript.

We believe that we have addressed all the reviewers’ comments by providing clarification of the writing as well as by streamlining the manuscript to provide a clear reflection of our work in the current version. A detailed point-by-point in annotated summary follows below. We fully hope that the manuscript is now of suitable quality for publication.

Reviewer 2

This paper needs major revision before publication.

1- There is a too lengthy introductory portion at the start of Abstract. Reduce it to maximum of two sentences.

According to the reviewer’s comment, the text has been modified.

2- Write Abstract in past tense.

According to the reviewer’s comment, the text has been modified.

3- Last paragraph of Introduction is highly confusing . If it is the work of any other scientist then it should be with a reference. If it is a part of the present study then it should be deleted from Introduction.

According to the reviewer’s comment, the text has been modified.

4- Add aims of the present study at the end of Introduction.

According to the reviewer’s comment, the text has been modified.

5- In Figure 1 A, standard error on the last bar is too high. Please check the raw data. It seems that values in one replicate are markedly different from the values in other replicates. If so, adjust this value by taking the mean of other replicates. Then apply statistical analysis of the whole data of this figure that will be different from the present one.

As reported in the manuscript, each reported value represents the mean ± SD of measurements carried out in triplicate and obtained from three independent experiments, thus in the figure is reported the obtained value and its variability. We can't explain the reason for the higher variability of this measurement and at the same time, we cannot change it.

6- In Figure 3, put * in the centers of the bars.

Done

Statistical analysis in Figure 5 seems highly confusing. Apparently it seems wrong. Bars with same height have different letters and those with different heights have same height. Recheck please.

We checked it and we can confirm that the reported statistical analysis is correct.

7- Format refences uniformly and correctly. Some references are incomplete. Write scientific names in Italics. Write names of journals in abbreviated form.

Done

8- Improve formatting throughout the paper. Especially check the chemical formulas.

Done

Round 2

Reviewer 2 Report

Authors revised the manuscript incompletely. This revision cannot be accepted.

1- Abstract still contain unnecessary long introductory portion. What is its need here. There is a separate Introduction in the paper. In Abstract, just give your findings, better to star from the aim of this study and completely delete the introductory portion that is more than 1/3 of abstract.

2- Last portion of Introduction is still written in a non-scientific way. Two paragraphs are written on aims and objectives. There are repetitions. Combine these aims and objectives in one paragraph.

3- In Fig. 1A, the large SD value at the last bar is due to one very large or very short value of one replicate. Please check again. If so, delete that extra ordinary different value and calculate means of the remaining replicates.

4- In Figure 5, to avoid confusion, authors should redraw the figure to group root and shoot bars in separate groups. 

Author Response

Authors revised the manuscript incompletely. This revision cannot be accepted.

We are really disappointed to read this comment, since we had responded to all the comments made, reducing the abstract, changing the last part of the introduction, and so on. The only thing we had not changed was the error bar in Figure 1A, which we have changed in this version (although we consider this request scientifically incorrect) since it does not change anything statistically.

1- Abstract still contain unnecessary long introductory portion. What is its need here. There is a separate Introduction in the paper. In Abstract, just give your findings, better to star from the aim of this study and completely delete the introductory portion that is more than 1/3 of abstract.

Abstract has been further reduced and now the introductory part is about 1/5 of the total text (41 words of 201 words). We fully hope we have met your request.

2- Last portion of Introduction is still written in a non-scientific way. Two paragraphs are written on aims and objectives. There are repetitions. Combine these aims and objectives in one paragraph.

We don't understand the meaning of the sentence “in a non-scientific way” and we find it rather offensive, anyway we changed again this part and again we fully hope we have met your request.

3- In Fig. 1A, the large SD value at the last bar is due to one very large or very short value of one replicate. Please check again. If so, delete that extra ordinary different value and calculate means of the remaining replicates.

The error bar in Figure 1A has been changed to meet your request, but as you can see it does not change anything statistically.

4- In Figure 5, to avoid confusion, authors should redraw the figure to group root and shoot bars in separate groups. 

This is a news with respect to the previous revision, in which you asked us to check statistical analysis and we did it; of course we are prepared to change the plot, but at the moment we do not consider it necessary since the two tissues analyzed are shown in different colours and we have grouped the two different S-treatments (S and L).

We hope you can accept our view.